# From Conventional to Craft Beer: Perception, Source, and Production of Beer Color—A Systematic Review and Bibliometric Analysis

**DOI:** 10.3390/foods13182956

**Published:** 2024-09-18

**Authors:** Nélio Jacinto Manuel Ualema, Lucely Nogueira dos Santos, Stanislau Bogusz, Nelson Rosa Ferreira

**Affiliations:** 1Postgraduate Program in Food Science and Technology, Institute of Technology, Federal University of Pará, Belém 66075-110, Brazil; lucelynogueira@gmail.com; 2Department of Agriculture Science, High School of Agriculture Science, Save University, National Road No. 1, Parcel No. 76, Chongoene 1200, Mozambique; 3São Carlos Institute of Chemistry (IQSC), University of São Paulo (USP), São Carlos, São Paulo 13566-590, Brazil; stanislau@iqsc.usp.br; 4Institute of Technology, Faculty of Food Engineering, Federal University of Pará, Belém 66075-110, Brazil

**Keywords:** beer coloring, bibliometry, Rayyan, VOSviewer, special beer, brewing, malting

## Abstract

Beer is a popular beverage consumed globally, and studies have emphasized the benefits of moderate consumption as well as its sensory effects on consumers. Color is a crucial sensory attribute, being the first aspect a consumer notices when assessing a beer’s quality. This review seeks to offer detailed insights into how brewing methods, raw materials, and the chemical diversity of beer influence the production of beer color. The chemical mechanisms responsible for color development and how consumers and color systems perceive the color of beer were assessed. A systematic review following the PRISMA methodology, coupled with a bibliometric analysis, was performed using (Rayyan 2022) and (VOSviewer 1.6.20) software to assess and evaluate the scientific research retrieved from the Web of Science Core Collection. The findings highlight the significant roles of malt types, heat brewing processes, control of chemical parameters, and innovative brewing techniques in conventional beer color production. Novel chromophores like perlolyrine, pyrrolothiazolate, and furpenthiazinate are thought to affect Pilsen-style beers, along with melanoidins, Strecker aldehydes, and 5-hydroxymethylfurfural (HMF) in conventional beers. In craft beers, such as fruit- or herb-based beers, flavonoids like anthocyanins, along with other natural pigments and synthetic colorants, are identified as the primary sources of color. However, studies related to the influence of chromophores like perlolyrine, pyrrolothiazolate, and furpenthiazinate on beer color are scarce, and emerging additives, such as pigments from microorganisms, spices, exotic herbs, and leaves of plants, on craft beer offer insights for future research.

## 1. Introduction

Beer ranks as one of the world’s most favored drinks. Following water and tea, it stands as the third most popular beverage and the most widely consumed alcoholic drink [1,2]. It is produced in most countries around the world, and China was the main producer in 2022, followed by the United States of America and Brazil [3]. The Czech Republic led consumption, with 144 L consumed per capita in 2019 and 188.5 L per capita in 2022 [3,4].

Beer is rich in nutrients, such as carbohydrates, amino acids, minerals, vitamins, and phenolic compounds, which combine through different chemical reactions to produce characteristic sensory attributes [5]. Flavor, aroma, and color are the main sensory properties that define its quality and acceptance. Among them, color is one of the most significant attributes as it creates the initial impression, attraction, and perception for the consumers [1,6]. In sensory evaluations, studies have indicated that beers displaying visually appealing colors are often preferred by consumers [7], showing that color is a vital element that informs many characteristics of the products we consume [8,9]. The color of beer contributes to the style and branding of the beer and is linked to its composition, as well as the technical and technological aspects of the brewing process [10]. A particular color can indicate the probable flavors and aromas found in a beer [11].

The range of conventional beer colors varies from pale straw, yellowish, golden, copper, amber, and brown to black. In craft beer, the color can achieve different hues depending on brewing proposes, influenced by specific additives or adjuncts [7,8]. Besides acting as colorants, the compounds that give beer its various colors, also help to protect light-sensitive elements and contribute to the beer’s preservation. They have nutraceutical properties, including antioxidant and antimicrobial activities, helping to prevent cancer, degenerative diseases, diabetes, and cardiovascular diseases [12].

According to the German Purity Law of 1516, beer is made only from barley, hops, and water, with the addition of yeast [1,5]. However, in an attempt to innovate beer styles, craft breweries are enhancing classic styles with locally sourced ingredients, leading to the use of uncommon additives or adjuncts such as fruits and other vegetable parts [13,14]. Besides the traditional offerings, beers produced with fruits, spices, vegetables, and other natural ingredients during the brewing process are gaining global popularity in response to the growing desire for novel drinking experiences, including visual stimuli [15]. In craft brewing, the addition of whole fruits, extracts, or parts of plants is one of the most economical and healthy ways to introduce colors and obtain unique characteristics in beer [16].

Achieving a good visual stimulus and maintaining reproducibility without compromising the beer’s originality poses a challenge, especially in craft breweries, due to the various factors that contribute to its production. Moreover, with limited research focused specifically on beer color production, this review aims to provide comprehensive information related to compounds and factors that contribute to the color of conventional and craft beer, as well as describe the impacts of brewing processes, raw materials, and chemical reactions involved in the production of beer color and highlight the magnitude of influence of each of the contributors. Bibliometric analysis was used as a comprehensive tool to identify research trends and knowledge gaps in this field.

As craft breweries are growing, the mechanisms and procedures indicated by several studies can contribute to the optimization of the color production process in this sector without affecting the original characteristic of the beer and inspire the traditional breweries.

## 2. Materials and Methods

The present systematic literature review was developed following the Preferred Reporting Items for Systematic Reviews and Meta-Analyses (PRISMA) checklist and registered in Open Science Framework, accessible at https://osf.io/ad2j7 (accessed on 6 September 2024) or at DOI https://doi.org/10.17605/OSF.IO/AD2J7.

### 2.1. Literature Search Strategy

The search strategy was based on the fundamentals of systematic review, which consists of identifying, selecting, and critically evaluating relevant research in an area of study, using well-defined criteria, whether or not using statistical methods to analyze and summarize the results of the studies [17].

The research was performed using the Web of Science (WoS) database (Core Collection), accessed from the CAPES portal between August 2023 and February 2024, using descriptors in English. The recommendations of the PRISMA statement (Preferred Reporting Items for Systematic Reviews and Meta-Analyses) for systematic review work were followed using a flowchart with four stages (identification, screening, eligibility, and inclusion) [17]. The research included documents published in the last twenty-one years (2004–2024), and the retrieved documents were ordered according to relevance. The guiding question for the research was: Regarding beer, what are the sources of color, and how is the color of beer produced and perceived? From this question, the main keywords used as research descriptors resulted. The research was conducted using the descriptors “beer AND color AND production OR beer color perception OR beer color” in the topic field of WoS (which includes the title, abstract, keywords, and keywords plus of papers indexed in the database). Boolean operators “AND” and “OR” were used as a research strategy for the relevant literature. The research was refined, including only reviews and articles as the document type. The documents retrieved from the database were exported for qualitative screening using Rayyan QCRI software, which is an application for web and mobile devices that helps streamline the screening and checking of keywords in titles and abstracts using a semi-automated process while incorporating a high level of utility in document analysis [18]. The documents selected in Rayyan were subjected to bibliometric evaluation, which is a quantitative method that uses mathematical data and statistical tools to measure the interrelationships and impacts of publications within a given area of research, resulting in a macroscopic view of a large number of academic literature [19]. The bibliometric evaluation was carried out using VOSviewer software (version 1.6.20), which is a program developed to build and visualize bibliometric maps based on network data, considering the strength of the connection between items [20]. The number of documents retrieved from WoS and the one included for bibliometric analysis were described in the PRISMA flowchart.

### 2.2. Eligibility and Exclusion Criteria

Studies that addressed aspects related to malting, types of malts, additives, and adjuncts in beer, use of fruits, herbs, spices, vegetable extracts, and the impact of brewing processes on beer color were considered. The exclusion criteria were based on the evaluation of the occurrence of the descriptors (keywords) used for research in the titles and abstracts of retrieved documents on the Web of Science. Other criteria used for exclusion were full-text availability and document duplication (carried out with the aid of Rayyan software). For this purpose, documents that presented up to 98% similarity were considered duplicates. The studies were limited to those written in English, and conference papers were also excluded.

### 2.3. Data Items and Defined Outcomes

For this review, the outcomes were grouped into two categories: Factors and parameters that influence the color in conventional beer (malting, Maillard reactions, caramelization, pyrolysis, and brewing-process-related reactions) and factors and parameters that influence the color in craft beer (natural vegetable pigments used in the production of fruit beer, synthetic additives in beer production, parts of vegetables used for the production of craft beer, and other ingredients that influence the color). Multiple Maillard reaction outcomes were reported, and we extracted the information that specifies the influence in beer, selecting the result that provided the most complete information for beer color issues. From the included studies, information related to the characteristics of the study matrix (beer) was extracted: EBC values, ingredients used, the production, and procedures applied to achieve the referenced EBC. For craft beers plus adjuncts or additives, the style of beer used as a base was taken into consideration. In conventional beers, the percentage of special malts used was also considered.

The significance of a study’s findings was assessed by evaluating the study design (methods and methodologies used to collect the data, duration of the brewing process, and production parameters).

### 2.4. Selection Procedures

Documents featuring the research keywords in their titles or abstracts were selected in the qualitative evaluation stage. The evaluation was performed by two independent authors (N.J.M.U. and L.N.S.) using blind evaluation in Rayyan QCRI software. The discrepancies were resolved by discussion to reach a consensus between the two authors, with a third author (N.R.F.) acting as a mediator if necessary. Those that only contained the keywords without available full-text or those with available full-text but with a corpus that did not meet the research objectives were excluded. Only documents that fulfilled the defined criteria were selected for quantitative evaluation in VOSviewer. The selection process in VOSviewer was based on bibliographic coupling measures. Documents with a minimum of five citations, and sources, authors, organizations, and countries with a minimum of five citations and three publications were selected (Figure 1). In this study, we considered keywords that had a minimum co-occurrence of five times in the titles and abstracts of the included papers. The documents that met the defined criteria in the quantitative evaluation in the VOSviewer software were selected to generate bibliometric maps. From eligible studies, the authors (N.J.M.U. and L.N.S.) manually collected the information by reading the outcomes. The third author (N.R.F) verified if the collected information matched the eligibility criteria.

## 3. Results and Discussion

The results are sequentially presented, beginning with the outcomes of the exclusion and inclusion process conducted in Rayyan and through full-text reading. Next, a description of the bibliometric evaluation results is provided in Section 3.2, Section 3.3, Section 3.4, Section 3.5 and Section 3.6. Subsequently, we present the findings reported in the selected studies, along with the corresponding discussion from Section 3.7, Section 3.8, Section 3.9, Section 3.10, Section 3.11, Section 3.12, Section 3.13, Section 3.14 and Section 3.15.

### 3.1. Quantitative and Qualitative Results

With the applied research strategy, a total of 1616 documents were retrieved, including 1577 (97.6%) original articles and 39 (2.4%) review articles. Three documents were detected as duplicates and were excluded, and 1613 were subjected to evaluation of titles and abstracts in Rayyan. A total of 1265 documents were excluded because they did not present the research descriptors in the titles and abstracts, and 348 advanced (Appendix A). The documents included at this stage underwent a full-text availability assessment, resulting in the exclusion of two that only presented a title and abstract, with the full-text not available on the Web of Science database. A total of 346 documents were thoroughly textually reviewed, and 74 were excluded due to their corpus not aligning with the objective of this review. Consequently, 272 documents were selected for the bibliometric analysis (Appendix A).

Figure 2 describes the flowchart with the steps of identification, screening, eligibility, and inclusion of documents found during the research, according to the recommendations of the PRISMA statement.

### 3.2. Bibliometric Analysis of Papers

Table 1 shows the ten most cited articles, as determined by the defined selection criteria in VOSviewer (a minimum of five citations). Out of the 272 articles included, the VOSviewer citations analysis identified 180 research papers and reviews that fulfilled the criteria. Notably, the study by Granato et al. [21] received the highest number of citations during the analyzed period, with 94 citations, followed by the work of Callemien and Collin [22], with 90 citations. The papers by Wannenmacher et al. [23], Ducruet et al. [24], and Bogdan and Kordialik-Bogacka [25], complete the top five, with 85, 80, and 67 citations, respectively. The articles that comprise the top five report the compounds, reactions, raw materials, and processes involved in beer color production, highlighting phenolic and Maillard reaction compounds. In the paper by Granato et al. [21], it was reported that the color of lager and brown ale beers is influenced by the type of malt and the malting process to which the production grains are subjected. These findings suggest that both the malt type and the malting process are determinants in beer color production. The top ten were completed by papers by Coghe et al. [26], Coghe et al. [27], Buzrul [28], Hellwig et al. [29], and Polshin et al. [30], with 67, 64, 63, 57, and 57 citations, respectively, which emphasize the influence of malt types, Maillard reaction, and hydrostatic treatment on beer color.

Table 1 shows that 60% of the top ten papers were published in the first decade of our analysis period (2004–2014), suggesting that the oldest papers tend to be the most cited. This trend has been previously reported by de Souza et al. [31], who attributed it to the fact that older documents have been indexed in the database for longer. The ten most cited documents were published in journals related to food science, except two, by the authors Coghe et al. [26] and Coghe et al. [27], which were published in the Journal of the Institute of Brewing, with a specific scope for brewing science, and one by Polshin et al. [30], which was published in the journal Talanta, with an analytical chemistry scope.

Regarding the type of document, 60% are research articles, contradicting Miranda and Garcia-Carpintero [32], who stated that review articles are those that receive the most citations in literature review research.

### 3.3. Bibliometric Analysis of Researchers and Organizations

Table 2 indicates the researchers and organizations that contributed the most publications and citations during the analysis period. It also shows the interrelations among researchers or organizations, represented by the total link strength. Total link strength is the strength of the co-authorship links of a given researcher with other researchers or the network of citations between two documents when they refer to at least one publication in common [20].

The bibliometric results from VOSviewer show that 1094 researchers from different countries and organizations participated in the production and publication of the 272 analyzed articles. Italian researcher G. Perretti from the University of Perugia published the highest number of documents, totaling 10, within the analyzed period. Among the top ten most cited authors, G. Perretti also received the most citations, with 143, which underscores his significant impact on the field of research. In addition to publications and citations, the value of total link strength of 1728 indicates that he is the researcher with the greatest collaboration and participation in publications in the area studied, linking him to several research groups. Among his collaborative works, several papers can be highlighted, such as that of Liguori et al. [33], which demonstrated the influence of the special malt content on the color of the beer. T. Becker and G. de Francesco, researchers from Tech Univ. Munich in Germany and Univ. Perugia in Italy, respectively, stand out next in the number of published documents, with six papers each. However, researcher T. Becker has a slight advantage in the number of citations, with 118 compared to 116 of G. de Francesco, which places him in second position in our ranking of most productive researchers.

The similarity and proximity in the number of documents among authors, from T. Becker, G. de Francesco, C. Spence, M. Gastl, O. Marconi to R. Prado (Table 2) suggest that research in this field is not dominated by a particular author or specific research group, indicating a widespread interest in this topic among various researchers and countries. Closer and higher values of total link strength indicate stronger relationships and greater collaboration between researchers. In this regard, researchers T. Becker and M. Gastl present total link strengths of 1373 and 1279, respectively, and both collaborated in the production of research in this field.

Germany registered the largest number of researchers, probably due to hosting one of the world’s oldest beer schools and being the top beer producer in Europe [3].

Regarding the most productive organizations, Katholieke Univ. Leuven in Belgium occupies the first position with 12 published documents, 445 citations, and a total link strength of 1613, pointing to its greatest contribution to scientific production in the study field and greater collaboration with other organizations, respectively. In the second position, Univ. Perugia in Italy stands out with ten published documents and 143 citations, driven by the publications of researchers G. Perretti and G. de Francesco. Tech Univ. Munich was the third most productive organization in the period under analysis, with the production of nine documents and 161 citations, and the authors T. Becker and M. Gastl stood out with the most contributions.

### 3.4. Country Bibliometric Analysis

Figure 3 represents the countries that contributed the highest number of papers and their respective interconnections. From the 272 documents analyzed in VOSviewer, 54 countries were identified, and 29 of them met the defined criteria (minimum of three documents and five citations). On the map, a circle’s size represents the number of documents a country has published, and a larger circle indicates a higher number of publications. Among 29 countries, Brazil, Italy, Belgium, Germany, and Poland are among the top five in terms of publication volume, with 31, 29, 27, 27, and 21 documents, respectively. Globally, Brazil is the third largest beer producer [3], which may justify the greater interest among several researchers to understand and improve the processes, parameters, and sensory properties of beer, which include beer color. Italy’s second position was influenced by the publications of researchers G. Perretti, G. de Francesco, and O. Marconi, who are among the top ten most productive researchers (Table 2).

Belgium, while being the third country in terms of published documents, holds the first position in the number of citations (708) and total link strength (4177). The high citation count is attributed to the fact that Belgium presented two organizations within the top ten for productivity, Katholieke Univ. Leuven, leading with 445 citations, and Catholic Univ. of Louvain, contributing 151 citations (Table 2). The total link strength is justified by the higher of the two organization’s total link strengths (1613 and 441, respectively). In turn, the higher total link strength supports stronger collaborations with Germany, England, and Italy, as evidenced by the thicker co-authorship connection lines between these countries (Figure 3). Additionally, Belgium, Germany, and England are renowned as the oldest beer schools globally, reflecting a shared interest in comprehending the brewing industry’s dynamics.

The slight similarity in the size of the circles indicates the proximity of the number of papers published by the top five, suggesting that research in this field is being carried out concurrently across different nations.

### 3.5. Bibliometric Analysis of Keyword Co-Occurrences

According to the bibliometric evaluation criteria, we considered keywords the items that appear more than five times in the included documents. Consequently, 1468 keywords were identified from 272 documents, and 93 of these met the defined criteria and were used in the construction of the map (Figure 4). The circles on the map represent the occurrence of keywords in the documents, and the larger the circle, the greater the occurrence of the keyword in the analyzed papers.

Figure 4 shows that the keyword beer has the largest circle, meaning that within the analyzed papers, it occurred most frequently, with barley and malt following. In the VOSviewer co-occurrence analysis, keywords linked by a line are considered to have a thematic connection, with shorter lines indicating a stronger relationship between them [34]. And in turn, the coloring on the map indicates the cluster where the keyword belongs [31].

The co-occurrence analysis of the keywords resulted in six clusters that are identifiable in Figure 4 by six colors. Cluster one with 21 items is represented by red, with the item malt being the most frequently occurring keyword, justified by the larger size of its circle in the cluster.

Cluster two, with 18 items, is represented by green, and the keyword with the highest occurrence is beer. Cluster three has 16 items, is delimited by blue, and barley is the keyword with the highest occurrence. Clusters four (15 items), five (12 items), and six (11 items) are represented by yellow, purple, and brown, and the most frequently occurring keywords are antioxidant activity, identification, and polyphenol, respectively. The greater occurrence of the terms barley, malt, polyphenols, and antioxidant activity within the clusters suggests that these terms constitute the main keywords guiding researchers in studies aimed at understanding beer color.

### 3.6. Journal Bibliometric Analysis

During the bibliometric analysis, 105 journals were identified, and 21 of them met the inclusion criteria (Figure 5).

The journal with the most documents published during the period under analysis was the Journal of the Institute of Brewing, with 31 publications and 619 citations (H-Index 61 and impact factor of 2.54 for the period 2022–2023), influenced by citations of papers by Coghe et al. [26] and Coghe et al. [27], which are among the ten most cited (Table 1), and Dugulin et al. [35]. The research by Coghe et al. [26] and Coghe et al. [27] contributed to understanding the impact of specialty malts on the color of the beer wort and the influence of the Maillard reaction on the beer color, respectively. Meanwhile, Dugulin et al. [35] suggested that the yellow hue in beer wort could be due to natural pigments like polyphenols and the vitamin riboflavin present in the malt.

The Journal of the American Society of Brewing Chemists occupied the second position, with 22 published documents and 295 citations (H-Index 45 and impact factor of 2.88 for the period 2022–2023). This position was achieved due to the contribution of several researchers, highlighting the work by Silva Ferreira et al. [36], who stated that the amber color of pale-colored Belgian beers is influenced by the polyphenol content (epicatechin and catechin) during storage, as well as by the dry hopping process.

The two periodicals have a scope linked to the manufacture and fermentation of beer, as well as raw materials and by-products of distillation. This likely justifies why they occupy the first two positions in the ranking. The third position is occupied by the European Food Research and Technology journal (H-Index 116 and impact factor of 3.93 for the period 2022–2023), closing the podium with 14 publications and 123 citations. The documents published by the three journals correspond to around 40% of the articles published by the 21 journals selected in VOSviewer and 43% of all citations, indicating the influence of these journals on the topic under study.

### 3.7. Perception of Beer Color

The world we live in is perceived through our five classic senses: vision, hearing, touch, taste, and smell. Each of these senses is interpreted subjectively by the brain [37]. Color is a sensory attribute perceived by vision and defines consumer’s expectations regarding the flavor, aroma, properties, and quality of food and beverages [8,9,11]. Regarding the expectations created by the color of the beer, it was reported that German brewing companies traditionally produce lager-style beers with low alcohol content, low bitterness, and light color, and therefore, German consumers associate light-colored beers with low bitterness. In the same way, in sensory evaluation, it was reported that consumers indicated that lighter beers quench thirst more than darker ones [38]. Carvalho et al. [11] reported that color influences consumers’ expectations of the taste and price of beer, with dark beers perceived as having a strong, bitter flavor, a fuller body, and a higher price (based on first visual contact). However, in the same study, the tasting experiences showed that there were no significant differences between light and dark beers in terms of flavor, indicating that color exerts an influencing power on the perception and expectation of the properties and characteristics of beers.

Therefore, in the beer industry, color is used as a marketing tool due to its influence on the interpretation of beer quality [39].

Color is not an inherent property but a perception in the mind of the observer. The perception of color in food involves three components: a source of light, the object (the food itself), and a detector (such as an eye or a diode). Additionally, the interpretation of color in liquid foods such as beer is a condition that depends on the composition or matrix of the food and the interpretation of the visible light that enters the eyes, whose electromagnetic spectral distribution varies from a wavelength of 380 to 780 nm and is modulated by the physical and chemical properties of the food matrix. In the eye, light comes into contact with the retina, which has rods and cones that translate the optical image into a pattern of nervous activity that is transmitted to the brain, and depending on the wavelength received and reflected, the brain will interpret the respective color [40,41,42]. The perception of color in humans is mostly trichromatic, defined by three types of opsin proteins expressed in cone neurons in the eye. The three types of cone receptors are categorized as red, green, and blue [8,43]. The presence of receptors in varying amounts explains why sensitivity to color varies among individuals, a phenomenon attributed to genetic differences. This accounts for the distinct perceptions of the same color by different consumers [40,43].

During the brewing process and in the final beer, color is assessed using different systems that approximate human eye perception, like CIE XYZ, CIE L*a*b*, CIE L*u*v*, RGB, and CIEDE2000 [42]. They present different distributions, as CIE XYZ determines the color on a three-dimensional color space based on the CIE color matching functions, while CIE L*a*b* is based on opponent color theory, L* shows the brightness, the position between light (values close to 100) and dark (values close to 0), a* is red (positive values) versus green (negative values), and b* is yellow (positive values) versus blue (negative values) [42,44]. The CIE L*a*b* system of chromatic coordinates is the most used tool for color perception in beer, especially in craft beer [41].

Numerous studies have shown that color can be characterized by three dimensions: hue, chroma, and lightness. Hue is how an individual recognizes an object’s color (e.g., blue, yellow, green, or red) as follows: 0° for red-purple, 90° for yellow, 180° for bluish-green, and 270° for blue. Chroma indicates the proximity of color to gray or a pure hue, and lightness denotes whether the color is perceived as light or dark. However, these three dimensions alone do not fully define the appearance of color. A comprehensive description of color appearance encompasses five perceptual dimensions: hue, chroma, colorfulness, lightness, and brightness [41,45,46].

### 3.8. Source and Production of Beer Color

The raw materials used in beer production, coupled with the technological process applied during brewing, allow the production of a huge variety of beer styles with different quality and sensory parameters [33]. The color of a beer is related to the beer composition and the technical and technological conditions of the brewing process [10].

Malt is a primary ingredient in beer production, serving mainly as a starch source and also contributing to the beer’s color and sensory attributes [47,48,49]. Shopska et al. [50] state that malt not only provides the necessary amount of starch for the production of fermentable sugars but also contributes to providing antioxidants, aroma, and the color of the final beer. Baigts-Allende et al. [51] registered values ranging from 4.16 to 38.52 SRM (yellow to black), with 55% of the beers showing a color between amber-deep amber/light copper, and they attributed this wide variation mainly to the type of malt used, brewing process, and storage conditions.

In conventional beers, color is derived, in large part, from malted grain [38]. Phenolic compounds (polyphenols) derived from barley husk or hops are secondary metabolites of plants and may also contribute to beer color [21,36,52,53]. Besides malt and polyphenols, some additives, such as caramel colorants added to the final formulation and syrups, also contribute to the color of beer [7,21].

The grains used in conventional brewing contain minimal concentrations of pigmented substances, and it is through the malting process that color is produced [40]. The stages of germination, kilning, and roasting in the malting process lead to the occurrence of the Maillard browning reaction, caramelization, and pyrolysis. These reactions generate a range of compounds that contribute to the color of both the wort and the final beer [21,54,55].

### 3.9. Malt and Malting Process

The malting process involves five steps: cleaning and assortment, steeping (early growth of the embryo), germination (formation of green malt), kilning (up to 80 °C) or roasting (final heat treatment 120–250 °C) of green malt, and screening, where the malt is separated from the rootlets and stems, which are the small roots and stems formed during germination (Figure 6) [47,56].

During the thermal processing of malt, a nonenzymatic browning reaction known as the Maillard reaction takes place. This reaction significantly influences the beer’s color and flavor through the formation of Maillard reaction products (MRPs) [56,57]. Besides the Maillard reaction, caramelization and pyrolysis reactions also occur during the malting and brewing process, leading to the production of coloring components in traditional beer [8,21,54,55]. The variety of malts produced can vary widely, depending on the thermal treatment used in the malting process and the length of the kilning or roasting. This range includes base malts like pale and pilsner malt, as well as specialty or dark malts. The color spectrum of these malts ranges from pale yellow to amber, brown, and nearly black (from 3 to 1600 EBC units), which later imparts color to the beer [27,56].

Gasior et al. [58] stated that the intention of using specialty malts is to color beer. Other authors reported that the main purpose of specialty malts is to improve some wort qualities such as pH, aroma, and color [26,48,50,56,59]. Intensely kilned or roasted dark malts give beer its deep red, red-brown, and mahogany colors [38]. Roasted dark malts are simultaneously responsible for the deep color of some beers and increased bitterness and aroma [49,56]. Specialty malts, also known as dark malts, are manufactured by a process similar to that of base malts but require higher kilning and roasting temperatures (110–250 °C). During specialty malt production, a group of compounds denominated Maillard reaction products are formed, leading to the specific color (EBC/SRM) of the produced malt and respective wort [47,56,58]. Specialty malts are divided into three types: color malt, caramel malt, and roasted malt [58].

Yang et al. [49] reported that chocolate malts (roasted malt) are usually applied in the production of Porter-style beers and are responsible for their dark appearance.

A study conducted by Liguori et al. [33] showed that the concentration of specialty malts used in beer production imparts the final EBC color. They found that as the percentage of caramel malt increased by 5%, the EBC unit color increased from 24 to 45, and if the percentage increased to 15% in the formulated recipe, the EBC unit color increased to 62, showing the influence of specialty malt in beer color production. In another study, the color of beer increased with the addition of 10% of Vienna malt, 10% of Munich malt, and 20% of melanoidin malt, from 4.99 EBC (pale malt beer) to 7.25 EBC (10% Vienna malt beer), 21.03 EBC (10% Munich malt beer), and 77.45 EBC (20% melanoidin malt beer), respectively [48]. The findings are aligned with the research conducted by Coghe et al. [60], which demonstrated that the color intensity of beer increases with the additional inclusion of specialty malt in mashing.

Dark beers such as schwarzbier, stout, Irish stout, and dunkles bock have been found to have higher EBC, lower L*, and higher a* values than pale beers due to the use of specialty malts which were kilned and roasted at higher temperatures [8]. Base malt (pale and pilsner malt) is the most common malt used for the production of beer worldwide and is yellow and straw-colored [33]. The most common color for a lager beer is pale amber or yellowish, which is produced by using base malt. A beer’s color can vary from light yellow with pilsner malt to a rich golden hue with pale malt, depending on the type of malt used [21]. Germany, for example, is known for its top-fermented wheat beers (weizen), pilsners, and weisse beers (beers that combine pilsner malt and wheat), all of which are light yellow due to the pilsner malt used in the brewing process [38].

Koren et al. [8] determined the color of 39 different types of beer and recorded low a* values (−3 to 4), high b* values (30 to 46), and high L* values (62 to 92) in Weisse beer and Czech pilsner as expected, which means that they are light to deep yellow, which corresponds to reality. Therefore, the registered color was attributed to the base malt used as a raw material in beer production. Another study reported a similar color shade with an a* value of −1.51, b* value of 18.31, and L* value of 94.92 for lager beer, while for brown ale beer, the a*, b*, and L* values were 35.75, 55.71, and 69.48, respectively. In this study, the difference was also related to the type of malt used. For lager beer production, only the base malt was used, and for brown ale beer, dark malt was added besides the base malt, which increased the beer color intensity, showing the apparent influence of the type of malt on beer color.

Dark beers are usually brewed from a pale malt or pilsner malt as the base, with a small proportion of darker malt added to achieve the desired shade. Very dark beers, like stouts, in addition to base malt, incorporate dark or specialty malts that have undergone extended roasting (such as black malt and chocolate malt) or roasted unmalted barley to achieve their distinctive color. Melanoidin malt, another specialty malt, is utilized to impart a deep amber to red-brown hue to the final beer [48].

### 3.10. Brewing Process in Beer Color

In addition to malting, the color of beer is also influenced by various stages of the brewing process, such as the heating steps (mashing, boiling, and pasteurization) and storage conditions [61].

#### 3.10.1. Mashing

During the mashing process, caramelized compounds are produced from the caramelization reaction of malt sugars, contributing to the beer’s browning index. Czech beers exhibit a deeper color compared to other European lagers, which is attributed to the traditional decoction mashing technique used in Czech lager production [8].

Mashing processes that restrict oxygen exposure and limit peroxidase activity lead to a decrease in color development during mashing due to enzymatic oxidation [2]. In addition to caramelization, the Maillard reaction takes place, producing tryptophan from malt peptides or proteins and 5-hydroxymethylfurfural from glucose or maltose originating from starch. These two substrates then combine to form perlolyrine, identified as a low-molecular-weight chromophore that influences the color of lager beer [62]. Besides enzymatic activity and the isomerization of alpha-acids, the pH level of water is a critical factor during mashing; a higher water pH typically results in a darker beer color [54,63].

#### 3.10.2. Boiling

Maillard reaction, caramelization, and oxidation of polyphenols occur during boiling, contributing to the color of the beer. The Maillard reaction is more frequent during boiling due to the high concentration of amino acids and carbohydrates present in the wort, leading to melanoidins production [64].

At this stage, further caramelization may occur, resulting in the development of color in the wort, which in turn determines the color of the beer, depending on the quantity of colored malt used. In traditional beer, the longer the boiling, the more color is picked up as melanoidin formation is promoted [33]. In the brewing of fruit beer, adding pulp at the start of the boiling stage can lead to darker beers due to the heightened extraction of polyphenols and various pigmented compounds [24]. Additional extraction of polyphenols derived from hops may take place during the wort boiling process when a brewer utilizes whole or pelletized hops, which contributes to the beer’s color through the leaching of flavonoids into the medium [40].

During the boiling of wort, some proteins break down and rise to the surface. If these proteins do not settle at the bottom of the kettle, they can remain in the beer, resulting in a haze that impacts the color [21].

#### 3.10.3. Storage

Over time, beer inevitably changes; haze and colloidal particles emerge, and its color darkens [2].

Particularly in the presence of oxygen and at elevated temperatures, beer color is primarily modified by processes involving enzymatic and nonenzymatic oxidation reactions, which lead to the oxidation of polyphenols and the production of melanoidins as a result of Maillard reaction, respectively [8,40]. These reactions result in a darker and less yellow hue, respectively, particularly in pale beers [8]. The discoloration of craft beer during storage has been attributed to the degradation of melanoidins and color pigments, primarily anthocyanins [24,65]. During storage, Gibson et al. [66] correlated the increase in aldehyde concentration with beer color modification, particularly in darker beers. De Francesco et al. [67] demonstrated that the longer the beer is stored, the greater the color intensity becomes, owing to the oxidation of polyphenols. They also found a correlation between the decrease in melanoidin content due to degradation during storage and the diminishing color of pale beers. Šavel et al. [68] investigated the aging of beers under aerobic and anaerobic conditions and observed an increase in brown color due to the oxidation of polyphenols such as ferulic and caffeic acid.

### 3.11. Primary Chemical Reactions Involved in the Production of Beer Color

#### 3.11.1. Maillard Reaction

The Maillard reaction is a nonenzymatic browning reaction where amino and imino groups at the N-termini and side chains of free amino acids, peptides, and proteins react with the carbonyl groups of reducing sugars, producing malt flavor and color compounds (MRPs) in the malting and brewing processes [48,59,69]. It involves three stages [55]. In the initial phase, the amino and imino groups undergo a reaction with reducing sugars, resulting in the formation of Amadori rearrangement products (ARPs), such as N-ε-fructosyllysine (FL) and N-ε-maltulosyllysine (ML). FL and ML are recognized as the most prevalent ARPs in malt and have been quantified in beer as well. These ARPs were identified to be the most abundant ARPs in malt and have also been quantified in beer. During the second phase, vicinal dicarbonyl compounds such as glyoxal, methylglyoxal, 3-deoxyglucosone (3-DG), 3-deoxygalactosone (3-DGal), 3,4-dideoxyglucosone-3-ene (3,4-DGE), and glucosone are produced from earlier-formed ARPs. In the second phase, the formation of dicarbonyl structures at the reducing end of di-, oligo-, and polysaccharides was observed, as well as the production of furfural and 5-hydroxymethylfurfural (HMF) from the dehydration of 3-deoxyglucosone (3-DG) and 3-deoxypentosone (3-DPs), respectively. Several 1,2-dicarbonyl compounds, among them the maltose degradation products 1-deoxymaltosone, 3-deoxymaltosone (3-DM), Strecker aldehydes, and pyrazines, have already been quantified in beer during this stage [47,48,58]. The last phase of the Maillard reaction involves the covalent modification of the N-termini and the nucleophilic side chains of amino acids, peptides, and proteins. Pyrraline, methylglyoxal-derived hydroimidazolone (MGH1), N-ε-carboxymethyllysine (CML), pyrraline 1a, formyline 1b, and maltosine are formed in the late stage of the reaction, and have been recognized as the predominant glycated amino acids present in beer [55,69]. Aside from glycated amino acids, low-molecular-weight compounds called chromophores and high-molecular-weight structures responsible for the dark color of heated protein/sugar mixtures, called melanoidins, are also formed [48,55,56,62,69]. In a study conducted by Hellwig and Henle [55], a novel Maillard reaction product derived from strecker aldehyde known as 5-(2′-formyl-5′-hydroxymethylpyrrol-1′-yl)-pentanal has been identified in dark malt. However, there are limited studies confirming its relationship with malt and beer color.

Gibson et al. [66] reported that an increase in strecker aldehyde was related to the color of dark beer during aging. Glycated amino acids derived from dark malt have been identified in both dark beer and bock beer [59], but there are no clear data that they directly impart the color of beer.

Therefore, the heightened pigmentation index of malt during the thermal process and the resultant color of the wort and beer are due to the formation of two groups of Maillard reaction products: low-molecular-weight chromophores and high-molecular-weight melanoidins. The low-molecular-weight chromophores have a minor role in color development, while melanoidins have the most significant impact on the color of beer [26,48]. The influence of melanoidins on beer color was confirmed in the evaluation of beer worts produced with different dark malt varieties [59].

Nagai et al. [62] isolated and evaluated the influence of low-molecular-weight chromophore on beer color and, as a result, found a yellow compound, which related it to pale beer color. The chromophore was identified as perlolyrine, which is a Maillard reaction product formed from tryptophan and 5-hydroxymethylfurfural or sugars [70]. Although its contribution to the total color of the beer was very low, it was observed that with increasing concentrations of perlolyrine in beer, the color intensity increased: pale beer (3.2–8.0 µg/100 mL) and dark beer (4.8–14.0 µg/100 mL) [62].

Pyrrolothiazolate, which is derived from the dehydrated isomer of 1-deoxyglucosone (1-DG) or thiol and amino groups of cysteine, has been identified as a low-molecular-weight chromophore from the Maillard reaction, exhibiting a yellowish color [71]. Noda et al. [72] isolated and identified another chromophore formed by acid hydrolysis of protein in the presence of xylose or by a reaction between cysteine and furfural under strongly acidic conditions, named furpenthiazinate, with yellow hues. However, these compounds have not yet been identified in beer, although reports suggest their presence in other beverages. Other chromophores correlated to pale beer have been reported to be 3-deoxyosuloses, 3,4-dideoxyosulos-3-enes, 3-hydroxy-2-butanone, hydroxyl-2-propanone, glycolaldehyde, glyoxal, and methylglyoxal [56].

The color of wort produced from pale malt and dark malt and the molecule’s weight were related using the gel permeation chromatography technique. The results indicated that the colorant compounds from pale wort (linked with the yellowish color of pale beer) were retained in the pores of gel beads and eluted, showing their low molecular weight (<10 kDa, elution at approximately 170 min) compared to high-molecular-weight melanoidin compounds from dark wort (>70 kDa, elution with the void volume after 70 min). This study suggested that the color of pale beer is related to low-molecular-weight colorant compounds, while the color of dark beer made from dark malts is related to high-molecular-weight melanoidins [27].

Melanoidin extracted by acetone precipitation (APE-M) and melanoidin extracted by macroporous resin adsorption (MAE-M) from dark beer were investigated. The results indicated that the melanoidin extracted by APE was nonpolar and of low molecular weight, while the melanoidin extracted by MAE had a high molecular weight [73].

Carvalho et al. [47] postulated that high-molecular-weight melanoidin may be produced through the polymerization of low-molecular-weight chromophores during roasting, as the yield of chromophores in black malt decreases concurrently with the increase in high-molecular-weight melanoidin. During malt roasting, melanoidin polymerization is related to the action of vicinal diketones and radical scavenging antioxidants since the formation of melanoidins coincides with the abrupt decrease in the level of vicinal diketones and radical scavenging antioxidants [27].

Besides high-molecular-weight melanoidins and low-molecular-weight chromophores, Gasior et al. [58] showed the correlation between 5-hydroxymethylfurfural (HMF), a byproduct of the Maillard reaction, and the rising browning index of malt, suggesting its potential impact on the color of the final beer. The authors evaluated the HMF content in commercial beers of different styles and observed that blond beers contain the range of 2.42–5.80 mg/L, amber beers 5.92–7.44 mg/L, and dark beers 6.29–7.52 mg/L, indicating that the HMF content increased as the browning index increased.

The colors produced by melanoidin in beer range from yellow, orange, and red, deepening to brown as the Maillard reaction progresses. Slightly kilned malts exhibit yellow shades typical of pilsner lager beers, whereas heavily kilned malts show amber and brown tones, indicative of British ales or Vienna lagers [40].

The content of Maillard reaction products is greatly affected by the temperature and duration of the heat treatment. Roasting processes, occurring at high temperatures (above 120 °C), are crucial for producing specialty malts, resulting in a higher accumulation of melanoidin and HMF content [50,58].

Pale and caramel malts are distinguished by their low-molecular-weight colorants, resulting from treatment at temperatures below 100 °C [50].

The yellow low-molecular-weight colorants (chromophores) are also produced in the mashing, boiling, and pasteurization stages of beer production, aside from kilning, as those stages occur at a temperature of around 100 °C [62]. To support that, Coghe et al. [27] showed that during intensive roasting, the lightness parameter (L*) consistently decreased. Conversely, the color shade parameters a* and b* peaked after 105 and 90 min, respectively. These results suggest that yellow-colored chromophores (b*) are predominantly produced before the compounds associated with a red color shade (a*). This indicates that the ideal temperatures for producing yellow Maillard pigments are lower than those for generating red compounds (125 °C to 155 °C).

#### 3.11.2. Caramelization

During the roasting stage and thermal brewing processes (prolonged mashing and boiling times), temperatures exceeding 120 °C and extreme pH conditions cause caramelization reactions. These reactions, due to the thermal decomposition of sugars without amino nitrogen, result in the formation of reddish and/or brown caramelization products [40,51,74].

Caramelization is a crucial nonenzymatic browning reaction that contributes to the flavor, caramel aroma, and brown color of thermally processed sugary foods [75]. The caramelization process occurs in different ways, firstly involving the equilibrium of anomeric and ring forms, the conversion of sucrose into fructose and glucose, condensation reactions, intramolecular bonding, the transformation of aldoses into ketoses, dehydration reactions, fragmentation reactions, and the formation of unsaturated polymers [76].

The heating source, equipment specifications, and brewhouse vessel configuration, along with extended boil times, elevated pH levels, and the use of high-gravity wort, are critical factors in determining the range of color in the final product [54]. Dicarbonyl compounds are part of the caramelization products found in malt and beer [55]. The rate of browning in malt or wort due to caramelization is linked to pH levels, as the reaction occurs slowly in acid pH and more rapidly in basic pH [40]. Caramelization is responsible for the production of specialty malts such as dark caramel malts and roasted malt used for the production of dark beers [77].

Regarding humidity, the color of specialty malt is developed during the later stages of caramelization at humidity lower than 5%, following 75 min of the process. The color formation during caramelization depends on time–temperature profiles, and optimal color production is achieved either through low temperature over an extended period or through high temperatures for a shorter duration [27].

#### 3.11.3. Pyrolysis

Above 200 °C, pyrolysis reactions predominate in the malting process, where carbon-carbon bonds are broken [56,77]. Pyrolysis is induced by thermal energy to generate products with a strong burnt flavor aroma and the production of intensely black pigments in malt that will impart the final beer color [40,54,56]. Pyrolysis reactions take place at elevated temperatures and are thus confined to the roasting stage, being utilized for the production of very dark malts, which are subsequently used to brew very dark beers [56,77].

### 3.12. Polyphenols

In conventional beer production, following the Maillard reaction, caramelization, and pyrolysis, the enzymatic oxidation of polyphenols from brewing cereal husks (barley, wheat, sorghum, oats, etc.) and hops vegetative matter serves as the secondary source of beer color. The oxidation of polyphenols yields a spectrum of colors from reddish-amber, which is noticeable in lighter beers [40], to yellow, red, and brown hues, depending on the specific chemical structure of the polyphenol compounds. These changes predominantly occur during mashing, runoff, storage, and aging [58,68,78].

The malt comprises 70–80% of beer polyphenols, while 30–20% comes from hops [15,23,78,79]. It is the phenolic compound oxidation during brewing that affects the beer color [2,51,52,67]. The content of esterified fractions of polyphenols increases during the germination, kilning, mashing, and boiling processes. However, it is primarily during storage and aging that polyphenols generate chromophores associated with beer coloring [50,58].

Color alterations from polyphenol oxidation become most noticeable in pale lager beers during storage, whereas in darker beers, these changes are concealed by the hues of colored and roasted malts [40].

Flavon-3-ols (catechins) monomers, proanthocyanidin oligomers, and ferulic acid are recognized as the primary polyphenol groups in both barley grain and malt. Conversely, epicatechin and sinapic acid are not found in barley grain, yet they are present in barley malt [8,52].

Higher-molecular-weight polyphenols contribute to the color of beer and to haze formation [77]. The main polyphenols responsible for the color of beer are flavonoids and phenolic carboxylic acids [80,81].

Polyphenols modify beer color through oxidation products of mono, di, and trimers of flavan-3-ols (combinations of catechins and epicatechin). Among the (epi)catechin monomer oxidation products, dehydrodi(epi)catechin has been identified as one of the compounds that influences beer during storage [36,52]. The main flavonoids that contribute to beer coloring (produced from barley) are flavan-3-ol monomers and proanthocyanidin oligomers [58,82]. Flavonols and flavones are yellow, whereas proanthocyanidin can be orange, red, blue, or purple [9,77,83]. The polyphenols that contribute to the yellowish color of pilsner are flavonols (kaempferol, myricetin, quercetin, rutin, hesperidin, and genistein) from malt and hops [16].

Flavonoids (proanthocyanidin) and phenolic carboxylic acid (ferulic acid) contribute to the amber color of beer [15,84]. Although in small quantities, the oxidation of catechins and epicatechins from hops and malt contributes to the brown color of dark beers such as Scottish heavy ale and Dunkel Weissbier [68]. A significant increase in the content of catechin and ferulic acid during the malt kilning process was reported, showing the relationship with the beer color intensity [58].

Martinez-Gomez et al. [79] reported that, generally, dark beers have the highest levels of total phenolic compounds (TPCs), demonstrating a correlation between the increase in TPCs and the color of the beer. Additionally, a strong correlation has been observed between EBC values and the contents of total polyphenols and flavonoids [81], suggesting that phenolic compounds significantly contribute to the color of beer. A similar correlation between the total polyphenol content in beer, its antioxidant activity, and EBC values has been reported [52,64,65,85,86].

Thiobarbituric acid from hops is indicated to react with carbonyl compounds in beer and produce a typical color at 530 nm [87].

Passaghe et al. [86] assessed a new dip hopping technique and observed higher EBC values in samples that utilized dip hopping compared to control samples. The infusion conditions might enhance the extraction of polyphenolic fractions that contribute to color.

### 3.13. Other Sources

Beer is a complex matrix where various factors interplay, and it is expected that, aside from Maillard reaction, caramelization, pyrolysis products, and polyphenols, other factors such as haze, vitamins, endogenous pigments, and water also influence its color [21,64,68,88].

Haze particles can affect the color appearance of beer, with proteins and polyphenols being the primary sources. Additional sources include processing defects, like fragments of filter media or adsorbents, and inorganic particles that form in the packaging, such as oxalates [21,52]. Pasteurization is known to accelerate colloidal haze formation and disrupts the stability between high-molecular-weight proteins and polyphenols, contributing to darker color [88]. In addition to haze, riboflavin, a yellow or orange-yellow substance from malt and produced by yeast cells during fermentation, is noted to contribute to the color of light-colored beers [35,40,64]. Moreover, natural pigments like carotenoids (carotenes and xanthophylls), chlorophylls (pheophytins), and their oxidation derivatives from brewing grains and hops, along with metal ions (which act as oxidation catalysts), significantly influence the color of light beers, as they present hues that vary from yellow to orange [46]. Yeast does not directly affect the color of beer; however, it can cause color loss by adsorbing colored substances onto its cell wall. Indirectly, yeast may influence beer color by causing turbidity in styles where it remains in the final product, like Hefeweizen [54,89]. Mikyška et al. [90] showed a decrease in the EBC value from 10 to 2 EBC, with a reduction in the number of yeast cells, and the same behavior was reported for the content of colloidal particles, associating them with an influence on beer color.

Šavel et al. [68] documented the use of malt extracts to adjust the final color of beer in the post-fermentation stage. In the same study, they investigated the effect of water type on the color of beer during brewing. It was observed that tap water influenced the beer’s color more than deionized water, and they attributed that observation to the presence of metals, a slightly elevated pH, and the buffering capacity of tap water. Copper, iron, and molybdenum have been identified as contributors to beer color due to their influence on haze formation and the oxidation of polyphenols [82,91]. Filtration or clarification of industrial beers by finning agents (isinglass and gelatin) and clarifiers confers low EBC values for the beers [13].

The amount of amino acids and reducing sugars available in grains, malt, and wort also contributes to color formation, as it has a direct influence on the formation of chromophores and high-molecular-weight colorant melanoidins [48].

### 3.14. Color in Craft Beer

Craft beers, including fruit beers and beer-based mixed drinks, are becoming increasingly popular [8,61,84]. Unlike conventional beers, special beers incorporate additional coloring components beyond caramelization, pyrolysis, and Maillard reaction products. Beer-based mixed drinks and fruit beers include a variety of dissolved colorants from fruits, spices, and additives or adjuncts that affect absorbance outcomes [8,92].

The use of other ingredients as adjuncts in addition to the traditional malt, hops, yeast, and water is a common practice in the preparation of innovative beer recipes [92,93]. Unmalted grains, fruit components (peels, roots, and pulp extracts), spices, flowers, and various vegetable parts are commonly used as additives [93]. The addition of fruits, vegetables, herbs, and other natural foods enriches the beer with compounds that contribute to the beer’s color [15,94].

In craft breweries, the production of beer with different colors from fruit or fruit by-products and vegetables is a common trend in the search for new beer styles with novel colors and flavors [93]. The most phenolic compounds that impart fruit-beer color are anthocyanins (cyanidins, malvinidins, peonidins, delphinidins, and pelargonidins) due to their high solubility and stability [1,13,84,95]. Most chromogenic substances in fruits are associated with anthocyanins [46]. The reddish hue of Lambic beers, specifically Kriek and Framboise, is attributed to the high levels of anthocyanins, particularly cyanidin, derived mainly from the fruits used in their production [15,84]. Essiedu et al. [13] produced beer supplemented with hibiscus, resulting in a deep red beer due to anthocyanins, delphinidin-3-O-sambubioside, and cyanidin-3-O-sambubioside. The addition of eggpeel extract (EPE) in pilsen beer has endowed the beverage with significant levels of anthocyanins, which accounts for the beer’s reddish hue. Among the five anthocyanins found in EPE, delphinidin-3-rutinoside shows the highest value [96]. In another study, Cioch-Skoneczny et al. [54] used grape pulp, marc, and must as adjuncts in beer production and obtained a reddish darker beer due to the influence of anthocyanins. Romero-Medina et al. [97] reported a high content of anthocyanins and an amber-red-cooper color in beer produced with pigmented corn. Lycopene was indicated to impart beer color produced from grapefruit [8], while carotenoids were noted to affect the redder color of fruit beer with persimmon juice [41]. In the brewing process using fruit pulp, the anthocyanins are mainly extracted during heating (mashing and boiling). However, extraction and thermal degradation of the anthocyanins from the fruits compete; in short boiling times, the content increases as a result of the extraction, and after 45 min of boiling, degradation becomes predominant [24,65,98].

The addition of omija fruits at the start of the boiling process led to a darker beer with a richer red hue, likely due to the development of brown Maillard reaction products [99]. De Brito et al. [1] observed a reduction in the color index of fruit beer produced with juçara pulp when heated to 100 °C for one hour. The study also noted a decrease in color when the pulp was added during mashing, attributing the color reduction to the adherence of the pigment to the malt bagasse that was discarded post-mashing. A similar result was observed in a sample produced with 20% of juçara fruit pulp added at fermentation, and the reduction in the color index was attributed to the adsorption of anthocyanin to the yeast cell walls, which were removed from the beer before the maturation stage [1,94].

The pulp of fruits is added before or during the process of fermentation, and the extract is added in the post-fermentation stages [46]. When fruit is added to beer before fermentation, the yeast does not metabolize the fruit’s natural pigments, which allows the beer to maintain the characteristic color of the fruit used [100]. Buiatti et al. [101] exploited the coloring properties of Friulan saffron to formulate a craft beer and found that higher saffron concentrations led to elevated EBC values, indicative of a deeper yellow hue in the beer produced.

The potential for incorporating cornelian cherry juices in brewing technology was investigated, with anthocyanins and flavonol derivatives identified as the compounds responsible for the red and yellow hues of the resulting beer [102].

### 3.15. Colorants

In craft beer, besides the endogenous pigments from adjuncts, it is a common practice to use color additives such as caramel color III or beer caramel, as well as synthetic colorants, for beer coloration [7,103]. Caramel is used as a colorant in the baking and beverage industries, serving to replace colors or compensate for color loss due to exposure to light, air, and temperature variations [104,105]. Caramel colorants have been classified into four classes (caramel color I, caramel color II, caramel color III, and caramel color IV) to satisfy the requirements of several food and beverage systems [103,105]. The process of producing artificial caramel coloring involves introducing a carbohydrate substrate into a reactor, where sugars are heated to enhance mixing, followed by the addition of a catalyst to initiate caramelization reactions. This occurs over several hours under specific temperature and pressure conditions. Once the desired color intensity is reached, the batch is quickly cooled, filtered, and stored [103]. In the brewing process, the caramel used to color dark beer is primarily the positively charged artificial caramel color III (E150c) [7]. This caramel is usually a dark brown to black with colors ranging from 2500 EBC to 15,000 EBC. The caramel compounds responsible for color are caramelan, caramelen, and caramelin formed during sugar caramelization [105]. The use of caramel coloring offers brewers a convenient means of ensuring consistent beer color without modifying beer flavor [39]. The color of pale lager beers exposed to light can be adjusted with caramel beer without affecting the flavor [106]. Carvalho et al. [11] used coloring agents RB7 and RB1500, selected for their minimal impact on flavor, to create two distinct beers based on color appearance, using blond-ale beer as the base. A higher concentration of the colorant produced a dark beer (50 EBC), while a lower concentration resulted in a pale amber beer (17.5 EBC). Fleet and Siebert [107] studied the incorporation of caramel color III at different concentrations and observed a darker brown hue at a higher concentration (4 mg/L) compared to a lighter shade at a lower concentration (2.5 mg/L). In addition to coloring agents and caramel color III, Stachová et al. [7] assessed the use of synthetic colorants in the dyeing of Easter herb-colored green beers sold in the Czech Republic. The study recorded the presence of yellow Tartrazine E102 (1.58–3.49 mg/L), Brilliant Blue E133 (0.45–2.18 mg/L), and Indigo Carmine E132 (2.36 mg/L). The authors noted that a mixture of blue and yellow dyes produces a green color, explaining the green hue of the beer analyzed. The edible, symbiotic, multicellular, filamentous blue-green microalgae cyanobacterium Arthrospira platensis (known commercially as spirulina) is used as a colorant in craft beer and is associated with green beer production [108]. Colorant wash blue, which is an iron powder solution embedded with a dye, was reported to be used to make green beer for a St. Patrick’s Day party [109].

## 4. Study Limitations

One of the limitations of this study is the scarcity of indexed and peer-reviewed research specifically addressing beer color, making it challenging to obtain and provide a comprehensive view of this subject. Most information in peer-reviewed scientific articles is often concentrated on malt color. Data on the use of color additives in beer were derived solely from a single study, which restricts the generalization of these findings. This scarcity renders the current research a pioneering effort in examining this significant sensory attribute of beer.

## 5. Concluding Remarks and Perspectives

This review offers a detailed examination of the latest research on the raw materials, technological processes, and compounds that influence the color of both conventional and craft beers, along with their possible effects. It was observed that the interaction of different compounds changes the beer’s medium. The choice of raw materials and the processes used determine the color profile of the final beer. The color spectrum in traditional beers is mainly influenced by the malt type, Maillard reaction products (low-molecular-weight chromophores and melanoidins), polyphenols, the brewing process, and water quality. On the other hand, craft beers frequently obtain their hues from anthocyanins, endogenous pigments of adjuncts, and additional synthetic colorants. Regarding color perception, many researchers indicated that beer color is best described using a three-dimensional chromatic system, especially when analyzing craft beers. The impact of melanoidin on beer color has been researched; however, studies that address the impact of chromophores produced mainly in the second stage of the Maillard reaction and novel chromophores like perlolyrine, pyrrolothiazolate, and furpenthiazinate on beer color are scarce.

This article also addresses challenges in producing beer colors from fruit, focusing on the optimal stage for adding ingredients. It notes that some color compounds extract best at moderate temperatures stages, while others require higher temperatures stages for optimal extraction, but quickly degrade when extensively exposed to these temperatures due to their sensitivity. Consequently, future research should focus on developing an improved method for achieving a high yield of colorants while ensuring their preservation throughout the brewing process. Concerning color perception, it has been observed that numerous studies have emphasized chemical and instrumental methods in color detection, with scant attention given to the sensory mechanisms of color perception by consumers and color perception through five perceptual dimensions. It has also been noted that there are limited studies addressing the chemical reactions involved in the color production of craft beers. Hence, future research should also focus on describing chemical reactions involved in the color production of specific craft beers and evaluating the effects of emerging additives, such as pigments from microorganisms, spices, exotic herbs, and leaves of plants such as Justicia secunda, on beer color.

## Figures and Tables

**Figure 1 foods-13-02956-f001:**
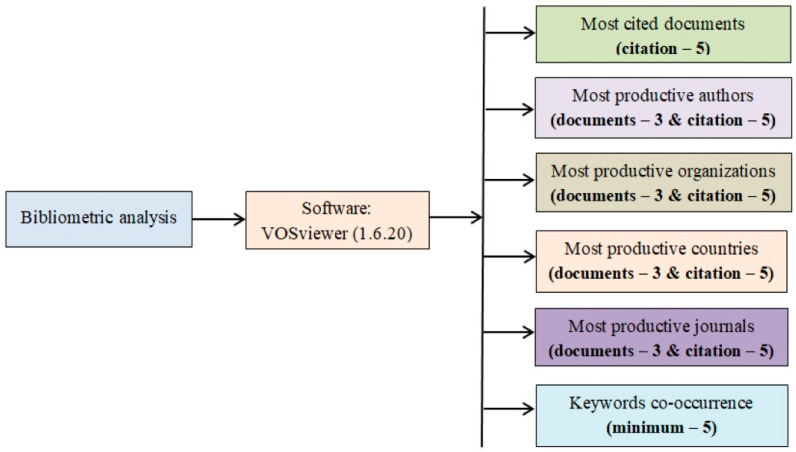
Flowchart describing the main steps and selection criteria used for bibliometric analysis: minimum citations, documents, and co-occurrence of keywords.

**Figure 2 foods-13-02956-f002:**
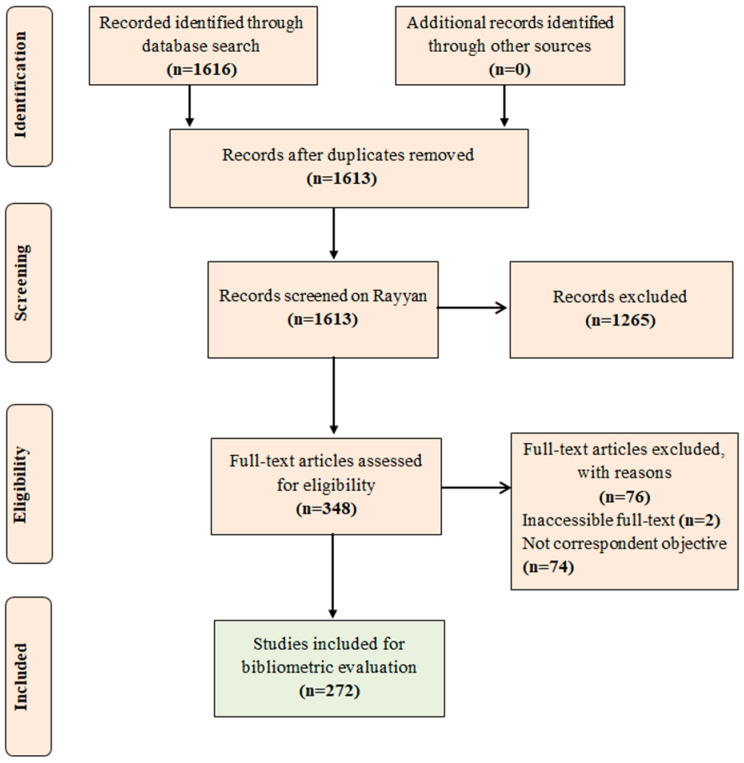
Flowchart showing the schematic representation of the methods of identification, screening, eligibility, and inclusion of documents in the research (Adapted from PRISMA).

**Figure 3 foods-13-02956-f003:**
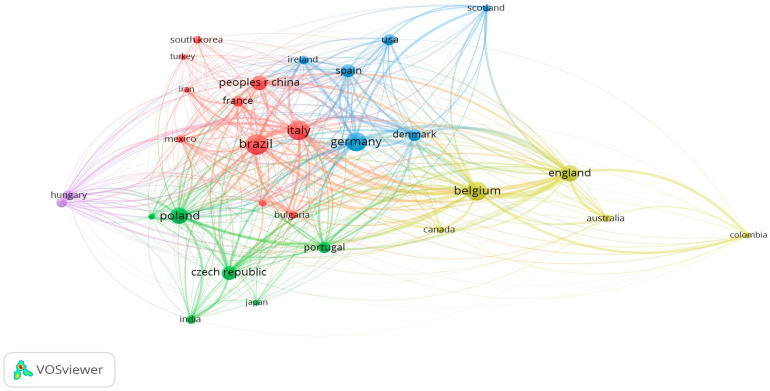
Network visualization map depicting the scientific production of 29 countries that meet the criteria outlined based on the 272 documents included in this study.

**Figure 4 foods-13-02956-f004:**
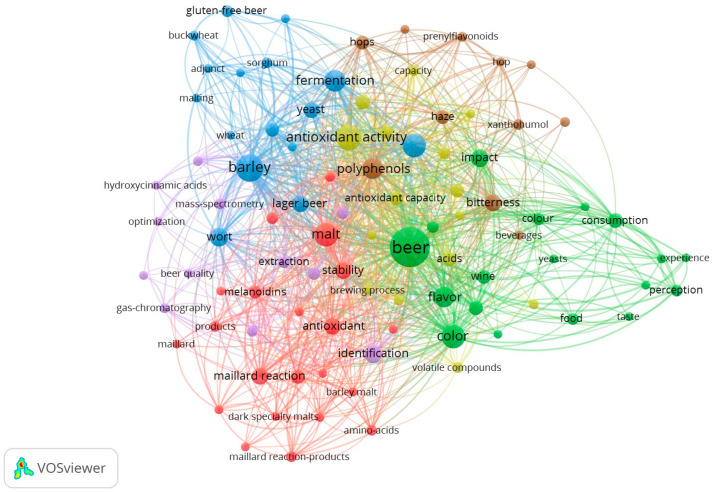
A network visualization map displaying the co-occurrence of all keywords within 272 documents that met the defined threshold for this study.

**Figure 5 foods-13-02956-f005:**
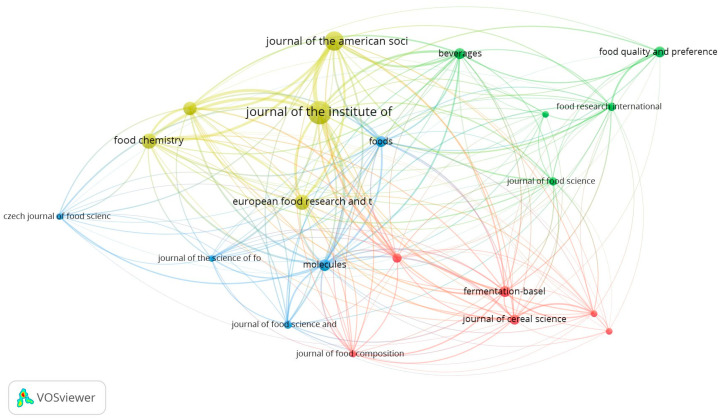
Network visualization map representing the most productive journals according to the volume of published documents.

**Figure 6 foods-13-02956-f006:**
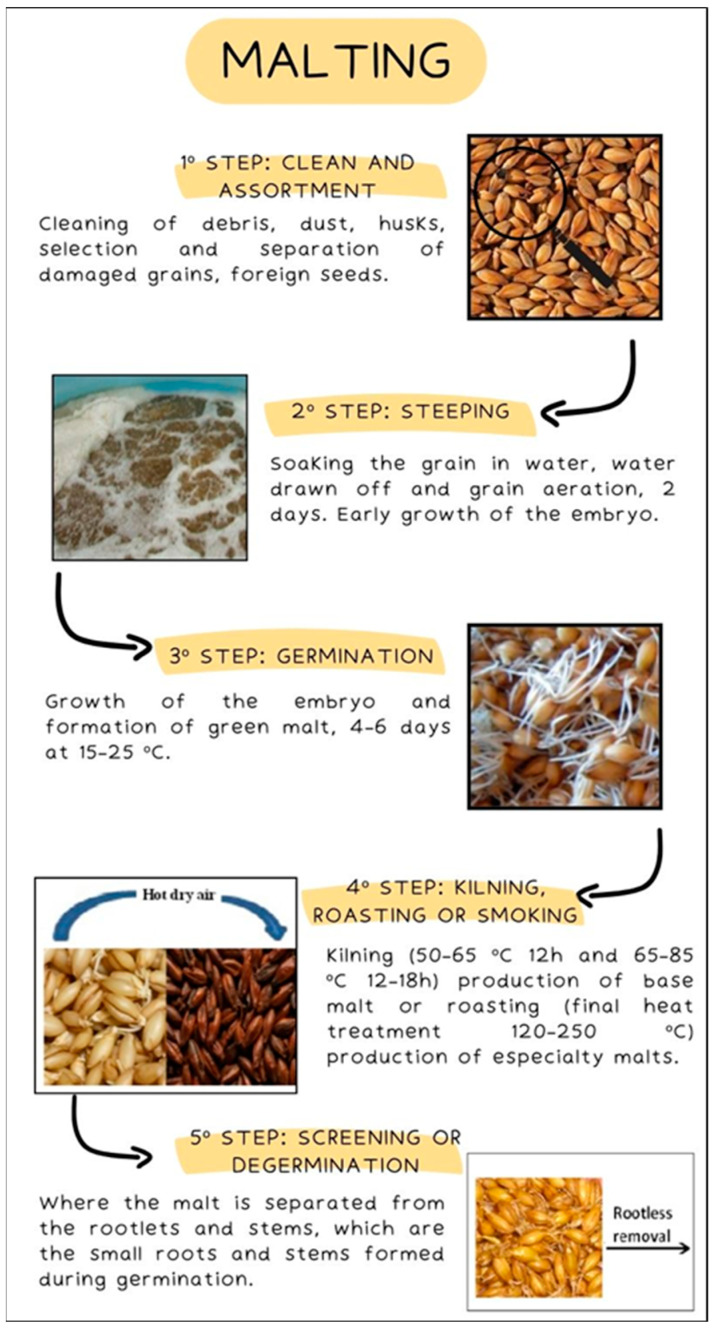
Illustration of main steps involved in the malting process, key events, conditions for occurrence, and objectives.

**Table 1 foods-13-02956-t001:** Most cited retrieved documents according to the criteria defined for bibliometric evaluation.

Ord.	Author/Year	Title	Journal	Cit.	ToD
1	Granato et al. [21]	Characterization of Brazilian lager and brown ale beers based on color, phenolic compounds, and antioxidant activity using chemometrics	J. Sci. Food Agric.	94	Article
2	Callemien and Collin [22]	Structure, Organoleptic Properties, Quantification Methods, and Stability of Phenolic Compounds in Beer-A Review	Food Rev. Int.	90	Review
3	Wannenmacher et al. [23]	Phenolic Substances in Beer: Structural Diversity, Reactive Potential and Relevance for Brewing Process and Beer Quality	Compr. Rev. Food. Sci. Food Saf.	85	Review
4	Ducruet et al. [24]	Amber ale beer enriched with goji berries—The effect on bioactive compound content and sensorial properties	Food Chem.	80	Article
5	Bogdan and Kordialik-Bogacka [25]	Alternatives to malt in brewing	Trends Food Sci. Technol.	67	Review
6	Coghe et al. [26]	Sensory and instrumental flavour analysis of wort brewed with dark specialty malts	J. Inst. Brew.	67	Article
7	Coghe et al. [27]	Development of Maillard reaction related characteristics during malt roasting	J. Inst. Brew.	64	Article
8	Buzrul [28]	High hydrostatic pressure treatment of beer and wine: A review	Innovative Food Sci. & Emerging Technol.	63	Review
9	Hellwig et al. [29]	Free and Protein-Bound Maillard Reaction Products in Beer: Method Development and a Survey of Different Beer Types	J. Agric. and Food Chem.	56	Article
10	Polshin et al. [30]	Electronic tongue as a screening tool for rapid analysis of beer	Talanta	56	Article

Cit.—citations; ToD—type of document.

**Table 2 foods-13-02956-t002:** Most productive researchers and organizations based on the number of published documents.

Ord.	Author	Documents	Citations	Total Link Strength	Country
1	Perretti, G.	10	143	1728	Italy
2	Becker, T.	6	118	1373	Germany
3	De Francesco, G.	6	116	1341	Italy
4	Spence, C.	5	123	263	England
5	Gastl, M.	5	106	1279	Germany
6	Marconi, O.	5	68	908	Italy
7	Cioch-skoneczny, M.	5	35	539	Poland
8	Kunz, T.	4	97	672	Germany
9	Collin, S.	4	53	558	Belgium
10	Prado, R.	4	21	702	Germany
**Ord.**	**Organization**				
1	Katholieke Univ. Leuven	12	445	1613	Belgium
2	Univ. Perugia	10	143	821	Italy
3	Tech. Univ. Munich	9	161	696	Germany
4	Agr. Univ. Krakow	8	94	533	Poland
5	Univ. Porto	6	125	320	Portugal
6	Catholic Univ. Louvain	5	151	441	Belgium
7	Univ. Oxford	5	123	618	England
8	Univ. Copenhagen	5	104	485	Denmark
9	Wroclaw Univ. Environm & Life Sci.	5	49	281	Poland
10	Cracow Univ. Technol.	4	14	328	Poland

## Data Availability

The original contributions presented in the study are included in the article/Appendix A, further inquiries can be directed to the corresponding author/s.

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
