# Peer review of "From Conventional to Craft Beer: Perception, Source, and Production of Beer Color—A Systematic Review and Bibliometric Analysis"

_foods, 2024, doi:10.3390/foods13182956_

Round 1

Reviewer 1 Report

Comments and Suggestions for Authors

The author attempts to comprehensively discuss the generation and changes of color substances in beer, which has positive significance for the study of beer quality.

However, the arrangement of subheadings in the Results and Discussion section of the manuscript can easily cause confusion for readers.

1.For example, in the Results and Discussion section, the secondary headings are sorted from 3.1 to 3.8. From 3.1 to 3.7, each section has a subheading, but there is no subheading for 3.8. It is illogical to start directly from 3.8.1 after the discussion in section 3.7.

2. Section 3.8.1 discusses malt and its manufacturing process, while 3.8.2 to 3.8.4 discuss three important chemical reactions. In section 3.8.5, separately discusses polyphenolic compounds, and 3.8.6 returns to the fermentation process of beer. The author repeatedly intersperses discussions between topics such as processes and chemical reactions under the same subheading, causing confusion and frequent content jumps for readers.

3. Section 3.8.8 mainly discusses the color of craft beer, and its content only includes the 3.8.8.1. Why set up this 4-level subheading?

Author Response

Comment 1: The author attempts to comprehensively discuss the generation and changes of color substances in beer, which has positive significance for the study of beer quality. However, the arrangement of subheadings in the Results and Discussion section of the manuscript can easily cause confusion for readers.

Response 1: After deep revision, we noticed it, so we are grateful for the observation.

Comment 2: For example, in the Results and Discussion section, the secondary headings are sorted from 3.1 to 3.8. From 3.1 to 3.7, each section has a subheading, but there is no subheading for 3.8. It is illogical to start directly from 3.8.1 after the discussion in section 3.7.

Response 2: We consider this observation important and we agree that the numbering should have started from point 3.8. for sure. There was a failure in the automated numbering; the subheading “Country bibliometric analysis” was not numbered, and we believe that this affected the rest of the order. However, in the new numbering of the subheadings, this aspect was taken into account, and the indication of subheading 3.8. can be seen on page 11 in the revised copy.

Comment 3: Section 3.8.1 discusses malt and its manufacturing process, while 3.8.2 to 3.8.4 discusses three important chemical reactions. Section 3.8.5, separately discusses polyphenolic compounds, and 3.8.6 returns to the fermentation process. The author repeatedly intersperses discussions between topics such as processes and chemical reactions under the same subheading, causing confusion and frequent content jumps for readers.

Response 3: We also agree with this observation and to ensure this logic, we reorganized the subheadings, placing in sequence, first the processes in subheadings 3.9. malt and malting, and in subheading 3.10. brewing processes and then chemical reactions in subheading 3.11. We removed the polyphenols in the middle of the sequence of processes and chemical reactions, which was moved to subheading 3.12. The corrections can be seen on pages: 12 subheading 3.9.; 14 subheading 3.10.; 15 subheading 3.11. and 18 subheading 3.12 of the revised copy.

Comment 4: Section 3.8.8 mainly discusses the color of craft beer, and its content only includes the 3.8.8.1. Why set up this 4-level subheading?

Response 4: We removed the 4-level subheading and due to the new numbering, the subheading 3.8.8. was turned to 3.14 (page 20 of the revised copy).

Reviewer 2 Report

Comments and Suggestions for Authors

Overall comments

This study aimed to provide a systematic literature review of the factors that determine the color of both conventional and craft beer. It also describes how brewing processes, raw materials, and chemical reactions affect beer color production, highlighting the significance of each contributor's influence.

Overall, this manuscript is a valuable contribution to the field, potentially providing new insights for addressing research gaps related to the factors that determine the color of both conventional and craft beer, with implications for implementing more suitable techniques.

However, some issues, particularly those related to improving the flow and relevance in the introduction, adding clarity and specificity to the results and discussion, and thoroughly addressing limitations and research gaps in the concluding remarks, require deeper attention.

Introduction

The introduction could benefit from an improved flow by refining transitions and reducing redundancy.

Research gaps have not been clearly identified, particularly regarding unexplored biochemical pathways in beer coloration.

Additionally, the relevance of this study should be more explicitly stated, highlighting its importance for both academic research and the beer industry.

Materials and Methods

This section is structured and comprehensive.

Results and Discussion

The "Results and Discussion" section could benefit from adding an introductory paragraph to guide readers and improve clarity.

Some results can be made more specific and quantitative, particularly by including more detailed numerical data.

Additionally, ensuring that the text is fully descriptive and independent of figures enhances the understanding.

While some results are compared with existing literature, not all key findings are thoroughly contextualized. Strengthening this section by comparing the results with previous studies would help better validate and contextualize the findings.

Concluding Remarks and Perspectives

It  is important to address the study’s limitations thoroughly.

Additionally, clearly articulate the remaining research gaps, emphasizing areas where further studies are needed to advance understanding and address unresolved questions.

Comments on the Quality of English Language

The quality of the English language in this manuscript is generally good. However, minor editing is recommended to improve clarity and flow in certain sections, particularly regarding the transitions within the introduction and the specificity of the results

Author Response

Comment 1: The introduction could benefit from an improved flow by refining transitions and reducing redundancy.

Response 1: Thank you for pointing this out, we agree to improve some paragraphs of the introduction. To address this concern, we removed the fourth period from paragraph 1, as we understood that it compromised the transition. We added the fourth paragraph highlighted in purple in the introduction, as we believed that it improved the connection between the separate paragraphs. We added some connectors painted in purple and deleted some words that seemed redundant and some that were repeated unnecessarily. We added the problem for the study in the penultimate paragraph highlighted in purple, as we believe that it improved the transition to the entry into the objectives sentence. We also added the relevance of the study in the last paragraph.    

Comment 2: Research gaps have not been clearly identified, particularly regarding unexplored biochemical pathways in beer coloration.

Response 2: Perhaps we were unfortunate in using the term biochemical pathways, we would like to point out which compounds react to produce the final compounds responsible for the final color in specific craft beers. We believe that the research gaps are identified in heading 5, page 22 on the revised copy with more details.

Comment 3: Additionally, the relevance of this study should be more explicitly stated, highlighting its importance for both academic research and the beer industry.

Response 3: We agree that this is an important point, so we addressed the relevance of this study in the last paragraph of the introduction of the revised copy, page 2.

Materials and Methods

Comment 4: This section is structured and comprehensive.

Response 4: Thank you, we appreciate the observation.

Results and Discussion

Comment 5: The "Results and Discussion" section could benefit from adding an introductory paragraph to guide readers and improve clarity.

Response 5: An introductory paragraph was added in the Results and Discussion section, highlighted in purple on page 4 of revised copy.

Comment 6: Some results can be made more specific and quantitative, particularly by including more detailed numerical data.

Response 6: We are grateful for the observation, however, we had difficulty understanding the spirit of the question and consequently it was difficult to answer it and for that, we apologize. Nevertheless, we are open to satisfying the question through a more simplified and specific request.

Comment 7: Additionally, ensuring that the text is fully descriptive and independent of figures enhances the understanding.

Response 7: We are grateful for the suggestion and understand the placement, but in this aspect, we believe that in systematic and bibliometric review this procedure is common, and we are aware that not everything common is correct, therefore, in order not to go off the rails, we think it is ideal to follow the same description. Below we indicate some research that supports our observation:

1 - https://doi.org/10.3390/foods12234239

2 - https://doi.org/10.3390/foods13020257

3 - https://doi.org/10.3390/foods12163058

4 - https://doi.org/10.1016/j.foodres.2023.113671

5- https://doi.org/10.1016/j.foodres.2022.111061

6 - https://doi.org/10.3389/fspor.2024.1413182

Comment 8: While some results are compared with existing literature, not all key findings are thoroughly contextualized. Strengthening this section by comparing the results with previous studies would help better validate and contextualize the findings.

Response 8: Regarding this statement, we agree that contextualizing the results helps in understanding, and we believe that this was achieved in our results. However, we were unable to clearly understand which paragraphs needed more attention in this regard. To avoid disrupting the flow of ideas presented in the results, we apologize for not answering this question, but we are willing to satisfy you with a more targeted indication.

Concluding Remarks and Perspectives

Comment 9: It is important to address the study’s limitations thoroughly.

Response 9: We appreciate the observation and hope that we have understood the reviewer's position. To respond, we have decided to define a specific heading for study limitation, which in the revised copy can be seen on page 22.

Comment 10: Additionally, clearly articulate the remaining research gaps, emphasizing areas where further studies are needed to advance understanding and address unresolved questions.

Response 10: We believe that this is a very important observation, hence, additional research gaps, further studies, and unresolved questions are highlighted in purple in concluding remarks and perspectives (heading 5), page 22 in the revised copy.

Reviewer 3 Report

Comments and Suggestions for Authors

From conventional to craft beer: perception, source and production of beer color – A systematic review and bibliometric analysis  

Nélio Jacinto Manuel Ualema et al.

The qualities of beer are described and experienced mainly by its abundant sensory effects that are initiated by the sense organs of consumers. The power of sight and the visual perception is often overlooked within the mainstream of beer dissemination and information posts.

This systematic review considers the influences of various stages in the brewing processes on the formation of beer colors by inspection of chemical mechanisms on color development coupled to PRISMA guidelines and comprehensive bibliometric analysis based on the Web of Science Core Collection. Obvious control by ingredients including malt types, heating interventions, and differing brewing techniques has been combined with elaborate detection and identification of novel molecules in conventional beers. Evidently, the colors of craft beers using herbs, fruits, and the likes are to a great extent derived from natural substances that add to the overall complexities.

The exploration assembles accessible facts and materials, but new knowledge unpublished until now has been incorporated as well. The methods used  to collect data are current, critical, confident, and reliable on the 272 documents that were selected for the bibliometric analysis. Figures 3, 4, and 5 present network visualization maps that are quite attractive for easy interpretation by readers. Fine work! In general, the presentation is uncomplicated and clear to understand.

The study presented, as evaluated by this reviewer, undeniably reveals contemporary indications and consciousness on original beer values by virtue of color differentiations. It may well be considered as an eye-opener for brewers on a global scale.

Comments on the Quality of English Language

The text reads well, but a skilled editing could improve the overall quality.

Author Response

Comment 1: The text reads well, but a skilled editing could improve the overall quality.

Response 1: To improve the quality of the English Language, the text was submitted to rapid analysis on Grammarly.

Round 2

Reviewer 1 Report

Comments and Suggestions for Authors

I think the current manuscript can be accepted.

Author Response

Comment: I think the current manuscript can be accepted.

Response: Dear reviewer 1, the authors of manuscript ID: foods – 3138228, would like to thank you very much for your comments and suggestions, for the time you dedicated to improving the document and for suggesting the acceptance of the manuscript.